# Mitochondrial Dysfunction in Cardiorenal Syndrome 3: Renocardiac Effect of Vitamin C

**DOI:** 10.3390/cells10113029

**Published:** 2021-11-05

**Authors:** Raquel Silva Neres-Santos, Carolina Victoria Cruz Junho, Karine Panico, Wellington Caio-Silva, Joana Claudio Pieretti, Juliana Almeida Tamashiro, Amedea Barozzi Seabra, César Augusto João Ribeiro, Marcela Sorelli Carneiro-Ramos

**Affiliations:** 1Laboratory of Cardiovascular Immunology, Center of Natural and Human Sciences (CCNH), Federal University of ABC, Santo André 09210-580, Brazil; raquels.ns21@gmail.com (R.S.N.-S.); carolina.junho@gmail.com (C.V.C.J.); karipan84@gmail.com (K.P.); wellington.caios@gmail.com (W.C.-S.); juatamashiro@gmail.com (J.A.T.); 2Laboratory BioNanoMetals, Center of Natural and Human Sciences (CCNH), Federal University of ABC, Santo André 09210-580, Brazil; pieretti.joana@gmail.com (J.C.P.); amedea.seabra@gmail.com (A.B.S.); 3Center of Natural and Human Sciences (CCNH), Federal University of ABC, Santo André 09210-580, Brazil; cesar.ribeiro@ufabc.edu.br

**Keywords:** CRS3, mitochondria, nitric oxide, mitochondrial dynamics, vitamin C

## Abstract

Cardiorenal syndrome (CRS) is a pathological link between the kidneys and heart, in which an insult in a kidney or heart leads the other organ to incur damage. CRS is classified into five subtypes, and type 3 (CRS3) is characterized by acute kidney injury as a precursor to subsequent cardiovascular changes. Mitochondrial dysfunction and oxidative and nitrosative stress have been reported in the pathophysiology of CRS3. It is known that vitamin C, an antioxidant, has proven protective capacity for cardiac, renal, and vascular endothelial tissues. Therefore, the present study aimed to assess whether vitamin C provides protection to heart and the kidneys in an in vivo CRS3 model. The unilateral renal ischemia and reperfusion (IR) protocol was performed for 60 min in the left kidney of adult mice, with and without vitamin C treatment, immediately after IR or 15 days after IR. Kidneys and hearts were subsequently collected, and the following analyses were conducted: renal morphometric evaluation, serum urea and creatinine levels, high-resolution respirometry, amperometry technique for NO measurement, gene expression of mitochondrial dynamic markers, and NOS. The analyses showed that the left kidney weight was reduced, urea and creatinine levels were increased, mitochondrial oxygen consumption was reduced, NO levels were elevated, and Mfn2 expression was reduced after 15 days of IR compared to the sham group. Oxygen consumption and NO levels in the heart were also reduced. The treatment with vitamin C preserved the left kidney weight, restored renal function, reduced NO levels, decreased iNOS expression, elevated constitutive NOS isoforms, and improved oxygen consumption. In the heart, oxygen consumption and NO levels were improved after vitamin C treatment, whereas the three NOS isoforms were overexpressed. These data indicate that vitamin C provides protection to the kidneys and some beneficial effects to the heart after IR, indicating it may be a preventive approach against cardiorenal insults.

## 1. Introduction

Cardiorenal syndrome (CRS) is well known as a pathological link between the heart and kidneys. CRS type 3, also known as renocardiac syndrome, occurs when an acute kidney injury (AKI) leads to cardiac alterations [1,2,3]. There are several mechanisms participating in establishing CRS type 3. Among them, mitochondrial dysfunction has been highlighted as being crucial in the development of both kidney and heart injuries [3]. 

Dysfunctional mitochondria are involved in the development of a diversity of heart diseases, such as cardiac hypertrophy and myocardial infarction [4,5]. Secondary to the heart, the kidneys contain a large quantity of mitochondria, which are mainly concentrated in tubular epithelial cells and are essential to a variety of renal functions, such as ATP-dependent electrochemical transports and acid–base balance [6,7]. In addition, impaired mitochondrial functioning leads to the death of tubular epithelial cells, and consequently tubular dysfunction [8,9]. In this context, an imbalance in mitochondrial quality control causes damage to biological systems and induces cell death, which is directly involved with the development of cardiovascular and renal diseases [10,11,12,13]. Mitochondrial injury can be observed by analyzing processes related to mitochondrial dynamics, such as fusion and fission, respirometry, and release of reactive oxygen species (ROS). Similarly, reactive nitrogen species (RNS), such as nitric oxide (NO) and nitrite, are also produced by mitochondria. Once released, NO is converted into nitrite (NO_2_^−^), nitrate (NO_3_^−^), and S-nitrosothiol [14,15,16].

In addition to endogenous antioxidant mechanisms, exogenous antioxidants have been highlighted to protect cells against ROS and RNS, such as tocopherol (vitamin E) and ascorbic acid (vitamin C) [17,18]. Vitamin C is a well-known and cheap antioxidant that is reduced to ascorbate in physiological conditions, its effector, and active forms in biological systems. There is some evidence indicating that between 40 to 80 µM of vitamin C avoids lipoperoxidation in humans, neutralizes ROS by donating electrons, reduces iron and copper, and maintains vitamin E levels. Several studies have indicated protective effects of vitamin C against cardiovascular and renal diseases [19,20,21,22]. 

Previous studies by our group have shown the efficacy of renal ischemia and reperfusion (IR) as an important CRS3 model. It has already been demonstrated that oxidative stress is modulated in this model at 8 and 15 days, both in the heart and kidneys [23]. In this sense, the aim of this study was to evaluate the beneficial effects of vitamin C as a treatment for CRS type 3, mainly acting in the cells in order to reduce the release of RNS, and also conserving mitochondrial functioning. 

## 2. Materials and Methods

### 2.1. Experimental Animal Protocol

C57BL/6J male mice 6–8 weeks old (22–28 g) were obtained from the Animal Facility of Federal University of ABC (São Paulo, Brazil). The animals were kept in temperature-controlled rooms (24 °C) with a 12/12 h light/dark cycle, and had free access to standard mice chow and water. All surgical procedures and protocols were approved by the Ethics Committee on Animal Use of the Federal University of ABC (no. 5593240919) in accordance with the National Council for Control of Animal Experimentation (CONCEA).

The renal ischemia and reperfusion (IR) protocol was previously described [24,25,26]. In summary, the animals were anesthetized (10 mg/kg of xylazine hydrochloride and 80 mg/kg of ketamine hydrochloride from Agribands do Brasil Ltda, São Paulo, was given in a single dose of 200 µL for each animal i.p.) and submitted to occlusion of the left kidney pedicle for 60 min using a microvascular clamp (DL Micof, São Paulo). Mice were divided into three groups (the estimated number for each group was 8): sham (animals that only have the peritoneal cavity opened without occlusion of renal pedicle); IR8d (animals that had the left renal pedicle occluded for 60 min, and reperfusion for 8 days); and IR15d (animals that also had the left renal pedicle occluded for 60 min, but reperfusion for 15 days). 

A treatment protocol along with the IR protocol was conducted with ascorbic acid (57 mg/kg/day; Synth Lab, São Paulo, Brazil) in drinking water [27,28,29], which was started in two different time periods after IR. In the early group, the IR mice (animals that also had the left renal pedicle occluded for 60 min) were treated for 8 days after IR (day 0 to day 8). After the treatment time, the mice were left for 7 more days of renal reperfusion without vitamin C, being euthanatized 16 days after IR. In the late group, the IR mice were treated from the 15th to 22nd days after IR, and were euthanatized on the 23rd day (Figure 1A). After the reperfusion and/or treatment period, the kidneys and heart were weighed and stored at −80 °C, tibia lengths were measured, and the blood was collected by puncturing the inferior vena cava. The blood was centrifuged at 4 °C at 10,000× *g* for 15 min to obtain the serum.

### 2.2. Biochemical Analyzes 

The renal function was evaluated by assessing urea and creatinine serum concentrations using two specific kits (Labtest Ureia CE 27 and Labtest Creatinina 35, Lagoa Santa, Minas Gerais, Brazil) to obtain a colorimetric estimation and respective absorbances at 600 nm and 510 nm.

### 2.3. RNA Isolation and Analysis of Gene Expression 

Total RNA was extracted using Trizol (Invitrogen, Waltham, MA, USA) according to the manufacturer’s recommendations. The tissues from all experimental groups, stored at −80°C, were lysed using Polytron (Kinematica AG, Malters, Switzerland), and 500 μL of Trizol was used for every half heart or 1/3 kidney and centrifuged with chloroform at 4 °C at 12,000× *g* for 15 min. The uncolored phase containing RNA was purified using isopropanol and ethanol 75% and eluted in RNAse-free water. After the quantification and confirmation of RNA integrity by spectrophotometry using nanodrop (Thermo Fisher Scientific, Waltham, MA, USA) and agaroses electrophoresis (18S and 28S bands integrity), a reverse transcription was conducted in a thermocycler (Mastercycler gradient, Eppendorf, Hamburg, Germany) using a High-Capacity cDNA Reverse Transcription Kit (Thermo Fisher Scientific, Waltham, MA, USA) and a volume corresponding to 2 µg of total RNA for each sample. At the end of reverse transcription, we obtained 20 µL of cDNA. Then, 10 µL of cDNA was diluted in 130 µL sterilized Milli-Q water, and the dilution was used for real-time PCR (qPCR) (Rotor-Gene Q, QIAGEN’s real-time PCR cycler, Qiagen, Hilden, Germany), in order to analyze gene expression, using Sybr Green (QuantiTect™ Qiagen, Hilden, Germany), during 40 cycles (95 °C for 15 s, 60 °C for 30 s, and 70 °C for 10 s per cycle). The primers were designed using Primer BLAST and BLAST NCBI (Table 1). The control thresholds were acquired, and the ^−ΔΔCT^ method was used to calculate changes between the groups, with cyclophilin A as a housekeeping gene. The fold changes were compared to the sham group. 

### 2.4. Protein Extraction and Nitric Oxide Species Measurement

Total protein was extracted from tissues using RIPA Buffer (50 mM of Tris Base, 1% Triton x-100, 0.5% of sodium deoxycholate, 0.1% of SDS, 500 mM of NaCl, and 10 mM of MgCl_2_) lysis assay, protease inhibitor (200 µL of PMSF, 2 µL of DTT, 1 µL of Leupeptina, 2 µL of Pepstatina, and 795 µL of distilled water) and sonicated in short ultrasonic pulses (Fisherbrand™ Sound Enclosure for Model 50 and 120 Sonic Dismembrator, Hampton, NH, USA), followed by centrifugation at 4 °C, 10,000× *g* for 15 min. The protein concentration in samples was determined by a BCA kit (Thermo Fisher Scientific, Waltham, MA, USA) in a microplate reader (Gen5, BioTek, Winooski, VT, USA). Next, protein homogenates were diluted to 1 µg/µL and 1.5 µg/µL, respectively, in sterilized Milli-Q water in order to analyze the nitrite and S-nitrosothiol. 

Total levels of NO were quantified by measuring nitrite (NO_2_^−^), nitrate (NO_3_^−^), and S-nitrosothiol contents by using the WPI TBR4100/1025 amperometric NO sensor (World Precision Instruments Inc., Sarasota, FL, USA) with the nitric oxide specific ISO-NOP sensor (2 mm) [30,31]. For quantification of S-nitrosothiols, a solution of CuCl_2_ (100 mmol L^−1^) was used, and for NO_2_^−^, a solution of potassium iodide (KI, 100 mmol·L^−1^) in sulfuric acid (H2SO_4_, (100 mmol·L^−1^) was used. Data were compared to a standard curve obtained with S-nitrosoglutathione (GSNO) (for S-nitrosothiol quantification) and NaNO_2_ (for NO_2_^−^ quantification). Aliquots of 100 µL of diluted protein were added to the vial containing 15 mL of an aqueous potassium iodide solution (0.1 mol L^−1^) in sulfuric acid (0.1 mol L^−1^) for NO_2_^−^ detection, or in an aqueous copper chloride solution (0.1 mol/L) for S-nitrosothiol detection. In the case of NO_2_^−^ quantification, the content of NO_3_^−^ in the diluted homogenates were firstly reduced to NO_2_^−^ by a cadmium-based nitrate wire (Innovative Instruments, Tampa, FL, USA). Then, total NO_2_^−^ and NO_3_^−^ levels were determined. All analyses were performed in triplicate. The data were compared to a calibration curve, and the results are expressed as pmol/g of protein. 

### 2.5. Evaluation of Mitochondrial Function

#### 2.5.1. Preparation of Mitochondrial Fractions

Heart and kidney mitochondria were isolated according to Mirandola et al. [32]. The final pellet was resuspended in isolation buffer (10 mM HEPES buffer, pH 7.2 containing 225 mM mannitol, 75 mM sucrose, and 0.1% bovine serum albumin (BSA, fatty-acid-free)), without EGTA at an approximate protein concentration of 5 mg·mL^−1^.

#### 2.5.2. Mitochondrial Respiratory Parameters Determined by Oxygen Consumption

The oxygen consumption rate was monitored in an Oroboros O2k respirometer (Oroboros Instruments, Innsbruck, Austria) in thermostatically controlled (37 °C) and constantly stirred incubation chambers (2 mL), in mitochondrial respiration medium (MiR05) containing the heart or kidney mitochondrial preparations (0.5 mg protein mL^−1^) supported by glutamate plus malate (2.5 mM each), prepared according to Gnaiger [33]. The respiratory parameters (state 3, state 4, and uncoupled) were stimulated by 1 mM ADP, 1 µg mL^−1^ oligomycin (ATP synthase inhibitor), and 1 μM CCCP (uncoupler), respectively. Oxygen fluxes were calculated using DatLab7 (Oroboros Instruments, Innsbruck, Austria) and expressed as pmol O_2_ flux s^−1^ mg protein^−1^. 

### 2.6. Statistical Analysis 

The GraphPad Prism 6.0 software program (GraphPad Software, San Diego, CA, USA) was used to perform the statistical analysis. Data are expressed as mean ± standard error of mean (SEM). One-way ANOVA was used to statistically analyze all groups followed by Tukey’s post hoc test; *p* < 0.05 was considered statistically significant. 

## 3. Results

### 3.1. Vitamin C Preserved Renal Morphology and Function after IR Protocol

It was already demonstrated that the CRS model induced by renal IR caused a decrease in left kidney weight, as well as an increase in renal dysfunction markers such as urea and creatinine after 15 days of reperfusion [24]. This was corroborated by the present study, since the left kidney weight was significantly decreased after 15 days of reperfusion (Figure 1B), urea levels were increased after 8 and 15 days of reperfusion (Figure 1C), while creatinine levels were only increased in the IR 15d group (Figure 1D). Moreover, both early and late vitamin C treatments prevented the increase in urea levels (Figure 1C), but only early vitamin C treatment prevented creatinine levels from increasing (Figure 1D), suggesting that creatinine is an acute marker of renal dysfunction. 

These data indicated that vitamin C minimized the impact of the primary injury (renal) in the CRS type 3 model, and that early treatment is promising when considering these parameters. 

### 3.2. Vitamin C Modulated Mitochondrial Dynamics and Functioning after IR Protocol

As mentioned previously, mitochondrial dysfunction is essential for the development of cardiac and renal diseases [34,35,36,37]. It is also known from previous studies by our group that this CRS model induced by renal IR is responsible of the development of cardiac hypertrophy, which leads to cardiac dysfunction due to CRS [24,38].

Thus, some genes, such as OPA1 and Mfn2 for mitochondrial fusion and Drp1 and FIS1 for mitochondrial fission, were analyzed in both kidney and heart tissues in order to assess the role of mitochondrial dynamics in the unilateral renal IR. As a result, OPA1 expression was found to be reduced in IR 8d animals in the left kidney, although non significantly (Figure 2A), whereas Mfn2 expression was decreased in IR8d and IR15d groups, indicating possible fusion impairment (Figure 2B). Both early and late treatments with vitamin C seemed to improve mitochondrial fusion, as they increased OPA1 and Mfn2 gene expression. Furthermore, Drp1 gene expression was increased when the treatment was started 15 days after the IR (Figure 2C). However, treatment FIS1 did not show any difference (Figure 2D). There was no alteration in gene expression of the of mitochondrial dynamics biomarker in the heart after IR without the treatment, but the late treatment with vitamin C seemed to stimulate both fusion and fission, as OPA1, Mfn2, and Drp1 gene expression were increased in comparison with the sham group. Interestingly, Drp1 was increased and FIS1 was decreased in the early treatment (Figure 2E–H). 

Mitochondrial functioning was evaluated by measuring the respiratory states 3 (ADP-stimulated), 4 (non-phosphorylating), and U (uncoupled) by oxygen consumption rates in mitochondrial preparations supported by complex I substrates (glutamate and malate). A significant reduction in oxygen consumption rates was observed in all respiratory states, indicating marked mitochondrial dysfunction in the kidney of both IR8d and IR15d animals (Figure 3A). Moreover, early and late vitamin C treatments were able to attenuate the reduction in uncoupled oxygen consumption (state U). Likewise, respiratory states 3, 4, and U were found significantly reduced in the hearts of mice 8 and 15 days after IR. Furthermore, early and late vitamin C treatments completely abolished the IR’s deleterious effects on all respiratory states, suggesting that vitamin C had a beneficial effect for cardiac mitochondria (Figure 3B). Considering the renal function data and the recent mitochondrial dynamic and function results, we not only assumed renal recovery, but also that the progression of CRS to the heart could be protected by vitamin C due to its role in mitochondrial function and dynamics.

### 3.3. Vitamin C Modulated the Expression of Nitric Oxide Synthases Isoforms and Avoided Increase in Nitric Oxide Levels

We evaluated NOS isoform gene expression, as well as measured NO production, by quantifying its stable end products nitrite, nitrate, and S-nitrosothiol by an amperometric method in the kidney and heart in order to evaluate nitrosative stress. No changes were found in eNOS and nNOS expression the left kidney of IR8d and IR15d mice, whereas iNOS expression was slightly reduced (Figure 4A–C). Interestingly, a marked increase in eNOS and nNOS expression was observed in all groups receiving vitamin C (Figure 4A,B). In contrary, iNOS expression was found to be reduced in groups treated with vitamin C (Figure 4C). The three NOS isoforms evaluated remained unaltered in IR8d and IR15d mice in the heart. However, both the IR early and IR late vitamin C groups showed a pronounced increase in gene expression of all studied NOS isoforms (Figure 4D–F).

The assessment of NO intermediates in the kidney demonstrated that nitrite/nitrate levels were increased after 15 days of renal reperfusion, and the IR early group showed the same increased levels. However, nitrite/nitrate levels in the IR late group were reduced (Figure 5A). S-nitrosothiol showed increased levels only with vitamin C treatment (Figure 5B). Furthermore, nitrite levels in the heart decreased in the IR15d groups, while S-nitrosotiol was similarly reduced in the same group. The group treated earlier demonstrated a considerable elevation of both intermediates (Figure 5C,D). These data indicated that vitamin C may modulate the NO release by NOS in different ways in the kidneys and heart. Finally, all the results are summarized in Figure 6, in which we elaborated a flowchart regarding the crosstalk between the kidneys and heart after IR protocol and after vitamin C treatment, and the mitochondrial behavior.

## 4. Discussion

In this study, we aimed to unveil the effects of vitamin C in both the kidneys and heart in a CRS type 3 model induced by unilateral renal ischemia, and to assess the protective effects of this antioxidant more specifically in mitochondria. Vitamin C is a well-known and cheap treatment, and during CRS type 3, it seems to play a reno- and cardioprotective role, given that: (a) it is capable of protecting the kidneys against IR injury when administered immediately after the renal insult; (b) it is able to modulate biomarkers of mitochondrial dynamics in both the kidneys and heart; (c) it improves the mitochondrial function, mainly in the heart; (d) it modulates nitric oxide production by constitutive NOS isoforms in the kidneys and heart; and (e) it increases constitutive NOS isoform expression in both tissues, but inversely decreases iNOS expression in the left kidney. 

In fact, a kidney injury and reparative process strongly involve an inflammatory process, extracellular matrix production, and hypertrophy of remaining nephrons [39]. The reparative process could fail according to the severity, period, and frequency of ischemia and reperfusion injury, and in turn progress from AKI to chronic kidney disease (CKD) [40,41]. There are several reports regarding this indicating the loss of kidney weight after renal IR and impairment in renal function, including previous reports from our group [32,42,43]. Similar to this evidence, a reduction in the left kidney weight and worse kidney function were also observed 15 days after IR, indicating that the reparative process failed. Some reports have shown the possibility of vitamin C exerting a protective effect after renal IR injury, as well as in the present data through the improvement in renal function and maintenance of left kidney weight, which led us to postulate that vitamin C may exert a protective renal function [25,27,44]. 

Mitochondrial dysfunction and oxidative stress are considered direct mechanisms among the pathophysiological mechanisms involved in CRS [3]. As observed, mitochondria oxygen consumption in the kidney was reduced after renal IR, similar to other reports [6,45,46]. Some studies have considered the presence of two different subpopulations of interfibrillar and subsarcolemmal mitochondria in order to investigate heart respirometry [47]. Studies have reported an impairment in oxygen consumption during two animal models of heart failure in analyzing only subsarcolemmal mitochondria and both subpopulations [48,49]. Other reports indicated the same decrease of mitochondrial oxygen consumption in the heart as a response to renal insufficiency [50]. 

Regarding the interplay between mitochondria and antioxidants, there are several potential therapeutic approaches in order to modulate mitochondrial ROS production during renal and cardiac diseases models such as MitoQ, ascorbic acid, SkQ1, and α-tocopherol conjugated to lipophilic triphenylphosphonium cation, which demonstrated improvements in renal function, focusing on setting a redox balance and increased cardiac oxygen consumption [5,36,51,52,53,54]. Considering these reports, our data reinforce that vitamin C provides beneficial effects to mitochondrial function in both kidney and heart tissues. 

Dysfunctions in mitochondrial dynamics have a central role in the development of AKI and cardiovascular diseases, since excessive fission and low fusion are detrimental to both tissues [9,55,56]. A large number of studies indicated that OPA1 and Mfn 2 may suffer post-transcriptional and translational modifications, reducing their expression in kidney diseases [57,58,59,60,61]. However, OPA1 gene expression remained unaltered in our model, whereas Mfn2 expression was significantly reduced, indicating an impairment of mitochondrial fusion. Our data showed that vitamin C may stimulate mitochondrial fusion, but the pathways involved are still unknown. 

The excessive fission mediated by Drp1 in renal diseases is well known in the literature [62,63,64]; however, our data did not show its overexpression. Unexpectedly, the mice treated late with vitamin C showed an elevated expression of Drp1, corroborating previous reports. FIS1 overexpression was expected after renal insult, as mentioned by some reports [65], but FIS was unaltered in our model. Moreover, both OPA1 and Mfn2 were reported to be decreased regarding cardiovascular disease [55,66,67,68,69], but in our study, both biomarkers were not altered in the nontreated vitamin C groups. Drp1 overexpression was also documented as a mediator of fragmentation in heart diseases and CRS [63,70,71,72]. In the same way, FIS1 expression was also reported to be elevated [73,74]. Conversely, both fission biomarkers were not found to be altered in our model. Curiously, both fusion and fission seem to be balanced with vitamin C treatment, once again showing the importance of this treatment during CRS3. 

In order to verify the NO levels in the kidney and heart, analyses of its bioavailability and the gene expression of the three isoforms of NOS were conducted. Different reports have stated that NO produced by constitutive NOS isoforms is essential to maintain cardiac and renal functions [75,76]. However, NO bioavailability was reported to be decreased during cardiac and renal insults, along with iNOS overexpression and depleted eNOS and nNOS expression, and with the presence of oxidative stress [77,78]. Conversely, NO levels in the left kidney in this study and in a previous study by our group were increased [23]. Similarly to this evidence, NO levels in the heart were decreased in both studies. The previous study also demonstrated that the iNOS isoform was overexpressed in the kidney and heart after the renal IR [23]. However, NOS isoform expression remained at the same level as in the sham group. Some reports showed the link between antioxidants and NOS, mainly eNOS, through preservation of BH_4_, stimulation of NO release, and protecting kidney and heart [19,20,79,80,81]. The same protection may be mediated by vitamin C treatment, because NO levels released from nitrite and S-nitrosothiol were increased in the early treated groups. 

In conclusion, our results demonstrated that vitamin C administered as early as possible provides protection to the kidneys and heart against renal IR injury. These data suggest that vitamin C can be a preventive approach to alleviate both the kidneys and heart against insults.

The present study had some limitations. (1) The treatment in drinking water needed to be carefully performed to ensure the same dose, every day of treatment; (2) the extraction and preparation of mitochondria took a lot of time, which required a very well-adjusted euthanasia, and therefore surgeries should be scheduled to make it possible to perform all experiments on all groups at the same time; (3) although some references in the literature used the protocol for administering vitamin C in drinking water, and we controlled water consumption and preparation of vitamin C daily, it was possible that there was a variation between animals/groups; and (4) a control group at day 23 could provide some information regarding comparisons with the late vitamin C group.

## Figures and Tables

**Figure 1 cells-10-03029-f001:**
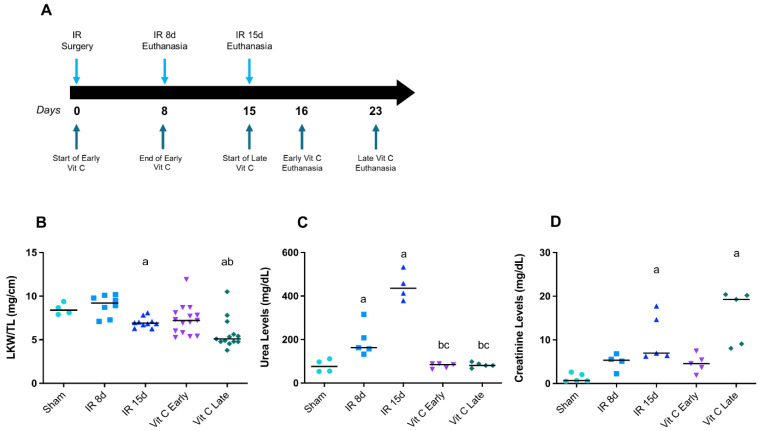
Effect of vitamin C on renal function after IR protocol. Experimental design of the ischemia and reperfusion (IR) protocol and treatments with vitamin C (**A**), left kidney weight/tibia ratio (mg/cm) (**B**), urea serum concentration (mg/dL) (**C**), and creatinine serum concentration (mg/dL) (**D**) in all different groups. Data are expressed as the mean ± standard error of mean (SEM), and one-way ANOVA followed by the Tukey’s post hoc test were performed. a vs. sham, *p* < 0.05. b vs. IR8d, *p* < 0.05. c vs. IR15d, *p* < 0.05.

**Figure 2 cells-10-03029-f002:**
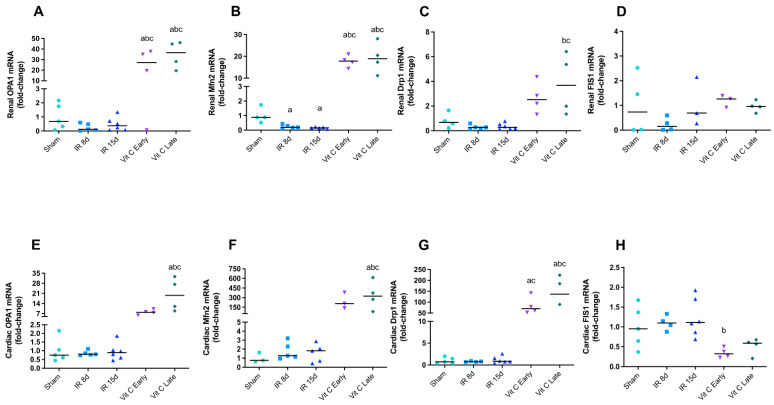
Mitochondrial dynamic components in CRS 3 model after vitamin C treatment. OPA 1 mRNA levels (**A**), Mitofusin 2 mRNA levels (**B**), Drp 1 mRNA levels (**C**), and FIS 1 mRNA levels (**D**) in the left kidney. OPA 1 mRNA levels (**E**), Mitofusin 2 mRNA levels (**F**), Drp 1 mRNA levels (**G**), and FIS 1 mRNA levels (**H**) in the heart. Data are expressed as the mean ± standard error of mean (SEM), and one-way ANOVA followed by the Tukey’s post hoc test were performed. a vs. sham, *p* < 0.05. b vs. IR8d, *p* < 0.05. c vs. IR15d, *p* < 0.05.

**Figure 3 cells-10-03029-f003:**
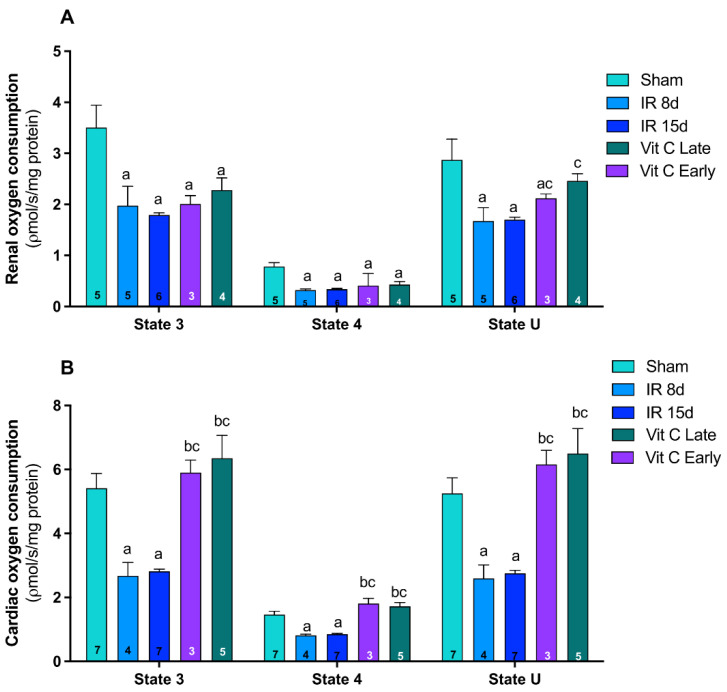
Mitochondrial function assessment by high-resolution respirometry in all different experimental groups. Respiratory states 3 (ADP-stimulated), 4 (non-phosphorylating), and uncoupled (U, CCCP-stimulated) were measured by oxygen consumption rates (pmol/s/mg/protein) in mitochondrial preparations from the left kidney (**A**) and heart (**B**) of mice. (**A**) Cardiac mitochondria oxygen consumption on states 3, 4, and U (pmol/s/mg/protein) (**B**). Data are expressed as the mean ± standard error of mean (SEM). One-way ANOVA followed by the Tukey’s post hoc test were performed for the selected pairs and number of animals within the bars. a vs. sham, *p* < 0.05. b vs. IR8d, *p* < 0.05. c vs. IR15d, *p* < 0.05.

**Figure 4 cells-10-03029-f004:**
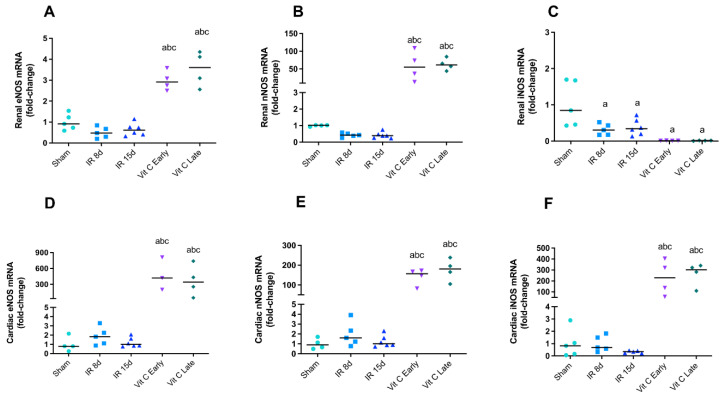
Vitamin C modulates nitric oxide synthase isoforms after IR. eNOS mRNA levels (**A**), nNOS mRNA levels (**B**), and iNOS mRNA levels (**C**) in the left kidney. eNOS mRNA levels (**D**), nNOS mRNA levels (**E**), and iNOS mRNA levels (**F**) in heart. Data are expressed as the mean ± standard error of mean (SEM) and one-way ANOVA followed by the Tukey’s post hoc test were performed. a vs. sham, *p* < 0.05. b vs. IR8d, *p* < 0.05. c vs. IR15d, *p* < 0.05.

**Figure 5 cells-10-03029-f005:**
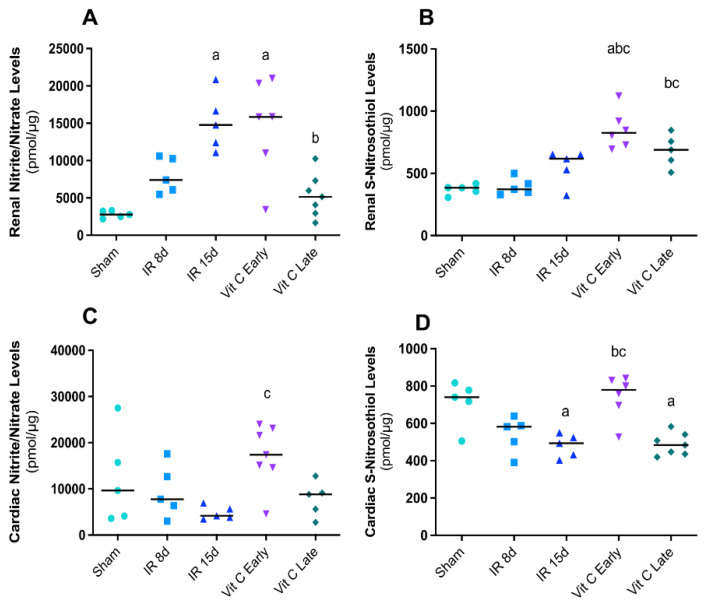
Impact of vitamin C on nitric oxide levels in CRS 3. Nitrite levels (**A**) and S-nitrosotiol levels (**B**) in the kidney. Nitrite levels (**C**) and S-nitrosotiol levels (**D**) in the heart. Data are expressed as the mean ± standard error of mean (SEM), and one-way ANOVA followed by the Tukey’s post hoc test were performed. a vs. sham, *p* < 0.05. b vs. IR8d, *p* < 0.05. c vs. IR15d, *p* < 0.05.

**Figure 6 cells-10-03029-f006:**
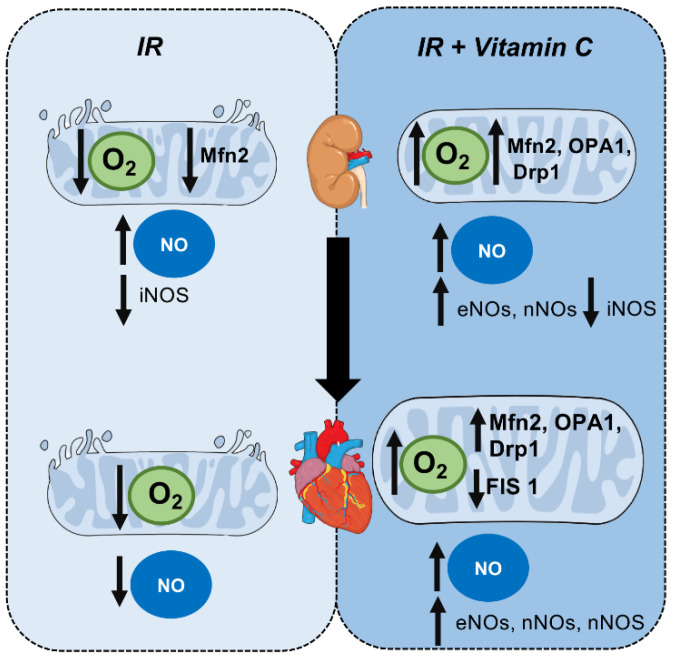
Crosstalk between the kidneys and heart after IR protocol after vitamin C treatment: mitochondrial behavior. Left side: effect of ischemia/reperfusion on mitochondrial function and nitric oxide levels. It is possible to observe a decrease in O_2_ production followed by an increase in nitric oxide levels in the kidney and a decrease in NO levels in the heart. Right side: effect of vitamin C treatment on mitochondrial behavior after ischemia/reperfusion protocol. After treatment with vitamin C (early or late), it was possible to note an improvement in mitochondrial function, modulation of mitochondrial dynamic genes, and a significant increase in NO levels in both the kidney and heart.

**Table 1 cells-10-03029-t001:** List of the primers.

Gene	Sense	Antisense
α-actin	GGCAAGATGAGAGTGCACAA	CGGAGAATGATGGTCCAGAT
ANF	ATCTGCCCTCTTGAAAAGCA	ACACACCACAAGGGCTTAGG
Cyclophilin A	AGCATACAGGTCCTGGCATC	AGCTGTCCACAGTCGGAAAT
nNOS	TCGATGCCAAGGCTATGTCC	CCTTGTAGCTCTTCCTCTCCTC
iNOS	GCTCTAGTGAAGCAAAGCCC	GGATTCTGGAACATTCTGTGCT
eNOS	CCCAGCCTCTCCAGCAC	GCCCATCCTGCTGAGCC
Mfn2	AGGTTGAGGTGACAGCGTTC	CTCCACCTGTCCAAGCTTCA
OPA1	TTAGAAAAGCCCTGCCCAGC	AGGTGAACCTGCAGTGAAGA
Drp1	GCCTCAGATCGTCGTAGTGG	TCCATGTGGCAGGGTCATTT
FIS1	CAGTGTTGCGTGTTAAGGGATG	TTCAAAATTCCTTGCAGCTTCGT

## Data Availability

The data is stored following the University’s data management policy and required by FAPESP.

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
