# Peer review of "Mitochondrial Dysfunction in Cardiorenal Syndrome 3: Renocardiac Effect of Vitamin C"

_cells, 2021, doi:10.3390/cells10113029_

Round 1
Reviewer 1 Report
The study conducted by Santos et.al is significant in terms of contributing to treatment options for CRS3.
Suggestions for the authors to consider;
- Grammar need to be looked at again before considering for publication, have highlighted and commented on a few in the paper. Its needs to be refined for a scientific audience (example line 30-33, 62)
- Provide more details on the dose of vitamin c chosen
- Provide details on number of mice in each group? did all survive?
- How many replicates were included in the ex vivo experiments?
- Why were two different study endpoints chosen (one has shorter IR injury time and the other has longer IR injury time)? would having similar length of IR injury with Rx at day 8 and day 15 have an effect on the findings?
- Data points should be shown on bar graphs.
- Original blots should be provided with the quantifications?
- PCR data should be presented using MIQE guidelines
- Discussion needs to be condensed and focused (see comment for line 304-306).
- State also the limitations of the study

Author Response
We would like to thank the Editor and the reviewers for their comments, corrections and suggestions that certainly improved our manuscript. All the points raised by reviewers were addressed. With these improvements, we hope that you will find our manuscript suitable for publication at Cells. Please, find below our answers to the suggestions/questions raised by the reviewers.
Reviewer 1
- The study conducted by Santos et.al is significant in terms of contributing to treatment options for CRS3.
Answer: We would like to thank for the comment regarding our manuscript.
- Suggestions for the authors to consider;
Grammar need to be looked at again before considering for publication, have highlighted and commented on a few in the paper. Its needs to be refined for a scientific audience (example line 30-33, 62)
Answer: We would like to thank for the suggestion and we’ve corrected the text in the new version of the manuscript.
- Provide more details on the dose of vitamin c chosen
Answer: The dose of vitamin c was decided, according to three studies, which used a non-invasive treatment protocols in animal models and they showed that the dosage of 57 mg/Kg/day in drink water was capable of protect kidney against ischemia and reperfusion injury and heart against left ventricle hypertrophy in hypertension [1, 2, 3]. These references were included in manuscript.
- Zhu, Y.; Zhanga, Y.; Zhanga, J.; Zhanga, Y. Evaluation of Vitamin C Supplementation on Kidney Function and Vascular Reactivity Following Renal Ischemic Injury in Mice. Kidney Blood Press Res. 2016, 41, 460–470, doi:10.1159/000443447.
- Bell, J.P.; Mosfer, S.I.; Lang, D.; Donaldson, F.; Lewis, M.J. Vitamin C and quinapril abrogate LVH and endothelial dysfunction in aortic-banded guinea pigs. Am J Physiol Hear. Circ Physiol 2001, 281, H1704–H1710, doi:10.1152/ajpheart.2001.281.4.H1704.
- Koul, V.; Kaur, A.; Singh, A.P. Investigation of the role of nitric oxide/soluble guanylyl cyclase pathway in ascorbic acid-mediated protection against acute kidney injury in rats. Mol Cell Biochem. 2015, 406, 1–7, doi:10.1007/s11010-015-2392-4.
Provide details on number of mice in each group? did all survive?
Answer: The experimental protocol was evaluated and approved by the ethics committee of the University, which defines, by mathematical formula, the number of animals allowed per protocol. In the present study, several different techniques were performed, which leads to a variation in the number of animals. For example:
- ischemia/reperfusion surgery: the mortality rate is low in this protocol (< 5%) but it is not zero, which means that in some sets of surgeries, we had the loss of some animals from some groups.
- High-resolution respirometry: the extraction of mitochondria is a long and laborious procedure, and we tried to keep the same time between extraction and analysis in Oroborus equipment. However, in some cases, the amount of mitochondria was not enough to carry out the experiment, decreasing the n in certain groups.
Finally, it is important to empathize that for each experiment it was necessary to prepare the samples according to the technique (for example, RT-PCR, NO measurement, etc.) considering the total amount of animals approved by ethics committee. Thus, sometimes, it was not possible to use the same tissue samples for all analysis proposed. Regarding the survival rate after surgery, we achieved a low mortality rate, <5%. The variation in number of samples is due to the limitation of the amount of tissue and different analyses performed.
How many replicates were included in the ex vivo experiments?
Answer: For Nitric oxide measurements and high resolution respirometry, we performed, at least, two replicates.
6- Why were two different study endpoints chosen (one has shorter IR injury time and the other has longer IR injury time)? would having similar length of IR injury with Rx at day 8 and day 15 have an effect on the findings?
Answer: These two different endpoints (IR 8 days and IR 15 days) were chosen in accordance with the previous studies of our group [1-4] where we characterized the renal and cardiac alterations. It’s important to empathize that after 8 days of reperfusion its possible to observe a systemic inflammation, electrical cardiac changes, redox unbalance, renin-angiotensin system activation and sympathetic nervous system activation. Besides, after 15 days of reperfusion, its possible to observe a concentric hypertrophy that was not accompanied by fibrosis. So, these two time points (8 and 15 days) are very important in these renal injury model.
- Cirino-Silva R, Kmit FV, Trentin-Sonoda M, Nakama KK, Panico K, Alvim JM, Dreyer TR, Martinho-Silva H, Carneiro-Ramos MS. Renal ischemia/reperfusion-induced cardiac hypertrop in mice: Cardiac morphological and morphometric characterization. JRSM Cardiovasc Dis. 2017 Jan doi: 10.1177/2048004016689440.
- Trentin-Sonoda M, da Silva RC, Kmit FV, Abrahão MV, Monnerat Cahli G, Brazil GV, Muzi-Filho H, Silva PA, Tovar-Moll FF, Vieyra A, Medei E, Carneiro-Ramos MS. Knockout of Toll-Like Receptors 2 and 4 Prevents Renal Ischemia-Reperfusion-Induced Cardiac Hypertrophy in Mice. PLoS One. 2015 Oct 8; 10 (10): e0139350. DOI: 10.1371/journal. Pone. 0139350.
- Silva W.C., Dias D.S., Junho C.V.C., Panico K., Santos R.N., Pelegrino M.T., Pieretti J.C., Seabra A.B., Angelis K., Carneiro-Ramos M.S. Characterization of the oxidative stress in renal ischemia/reperfusion-induced cardiorenal syndrome type 3. Biomed Research International, v. 2020, p. 1605358, 2020.
- Panico K, Abrahão MV, Trentin-Sonoda M, Muzi-Filho H, Vieyra A, Carneiro-Ramos MS. Cardiac Inflammation after ischemia-reperfusion of the kidney: Role of the sympathetic nervous system and the renin-angiotensin system. Cell Physiol Biochem;53(4):587-605. doi: 10.3594/000000159, 2019.
Data points should be shown on bar graphs.
Answer: We thank the reviewer for the suggestion. All data point were included inside the bars.
Original blots should be provided with the quantifications?
Answer: Not applicable.
PCR data should be presented using MIQE guidelines
Answer: Thank you for this observation and we include a detailed description of PCR methodology as suggested by MIQE guidelines
Discussion needs to be condensed and focused (see comment for line 304-306).
Answer: We appreciated the recommendation. The discussion in this sentence was corrected and condensed to make it focused.
State also the limitations of the study
Answer: The present study had some limitations. 1) The treatment in drink water needs to be carefully performed to ensure the same dose, every day of treatment, 2) The extraction and preparation of mitochondria takes a lot of time, which requires a very well-adjusted euthanasia and therefore surgeries schedule to make possible to perform all experiments of all groups at the same time, 3) Although some references in the literature use the protocol for administering vitamin C in drinking water, and we controlled water consumption and preparation of vitamin C daily, it is possible that there is a variation between animals/groups.
Reviewer 2 Report
This article by Raquel Silva Neres dos Santos et al, presents interesting findings in a mouse model of cardiorenal injury induced by unilateral renal ischemia/reperfusion. The authors report mitochondrial and NO alterations in the kidneys and hearts of mouse following a renal ischemic episode and the prevention of most of these alterations by Vitamin C treatment.
The main concern about the design of the study is that the untreated animals are studied at day 8 and 15 following IR, while the animals that received vitamin C were euthanized at day 16 and 23 after renal ischemia. This study design leads to a lack of a good comparison group, especially for the late vit C treatment group. In this case, the mice were studied at day 23, meaning that the kidney disease progression is more advanced as compared to mice with 15 days after renal ischemia, which is the group used to compare the effect of Vit C treatment. Thus, wrong conclusion with the late vitC group might be deduced. This might explain the lack of effect of vitamin C in some of the parameters studied such as creatinine, cardiac nitrate/nitrite levels or cardiac S-nitrosothiol levels.
How was the number of animals per group decided?
Minor:
What was the anesthesia method for the mice?
Even if the text mentions that Vit C treatment in the “early group” stops at day 8, it would be helpful to the readers if this is also graphically shown in Fig 1A.
Author Response
Reviewer 2
This article by Raquel Silva Neres dos Santos et al, presents interesting findings in a mouse model of cardiorenal injury induced by unilateral renal ischemia/reperfusion. The authors report mitochondrial and NO alterations in the kidneys and hearts of mouse following a renal ischemic episode and the prevention of most of these alterations by Vitamin C treatment.
The main concern about the design of the study is that the untreated animals are studied at day 8 and 15 following IR, while the animals that received vitamin C were euthanized at day 16 and 23 after renal ischemia. This study design leads to a lack of a good comparison group, especially for the late vit C treatment group. In this case, the mice were studied at day 23, meaning that the kidney disease progression is more advanced as compared to mice with 15 days after renal ischemia, which is the group used to compare the effect of Vit C treatment. Thus, wrong conclusion with the late vit C group might be deduced. This might explain the lack of effect of vitamin C in some of the parameters studied such as creatinine, cardiac nitrate/nitrite levels or cardiac S-nitrosothiol levels.
Answer: We would like to thank for the comment regarding our manuscript. The Vit C treatment was thought by the authors in these two specific time points given the possibility of a treatment starting in the begging of the syndrome - simulating a health patient that faces a renal IR due to some acute factors- and also - a patient that discovers the AKI and after starts the Vit C treatment. In both cases we have scientific evidences that supported us in theses choices.
For the early treatment: Previous studies form our lab have shown that after 8 days of reperfusion we have a strong participation of inflammation and ROS [1-6] and both components have maintained until the day 15 (another super studied timepoint), when we can find renal progression of AKI observed in mRNA levels of vimentin that is a tubular injury marker. In this sense, we have investigated an initial treatment based on known consequences for our model, which Vit C proved effective in preventing.
For the late treatment: Based in the same studies of implementation of renal injury after this model of renal IR, we have thought about an already established injury in the 15th day later treated with Vit C for some more days.
In fact, the observation of control animals in 23 days after reperfusion are primordial, however we have already performed the experiments of renal injury evaluation and reperfusion for 2, 4, 6 and 8 weeks and in our lab (not published). We have supported that the renal injury is maintained until the end of the 8th week compared with Sham 8weeks animals, demonstrating that the treatment with Vit C is important during the AKI protection.
- Silva W.C., Dias D.S., Junho C.V.C., Panico K., Santos R.N., Pelegrino M.T., Pieretti J.C., Seabra A.B., Angelis K., Carneiro-Ramos M.S. Characterization of the oxidative stress in renal ischemia/reperfusion-induced cardiorenal syndrome type 3. Biomed Research International, v. 2020, p. 1605358, 2020.
- Panico K, Abrahão MV, Trentin-Sonoda M, Muzi-Filho H, Vieyra A, Carneiro-Ramos MS. Cardiac Inflammation after ischemia-reperfusion of the kidney: Role of the sympathetic nervous system and the renin-angiotensin system. Cell Physiol Biochem;53(4):587-605. doi: 10.3594/000000159, 2019.
- Alarcon MML, Trentin-Sonoda M, Panico K, Schleier Y, Duque T, Moreno-Loaiza O, de Yurre AR, Ferreira F, Caio-Silva W, Coury PR, Paiva CN, Medei E, Carneiro-Ramos MS. Cardiac arrhythmias after renal I/R depend ON IL-1β, Journal of Molecular and Cellular Cardiology, 25. PII: S0022-2828 (19) 30083-5, 2019.
- Junho CV, Trentin-Sonoda M, Alvim JM, Gaisler-Silva J, Carneiro-Ramos MS. TLR4-induced cardiomyocyte hypertrophy is dependent of camkii through complement system and NF-KB. Brazilian Journal of Medical and Biological Research, 52 (7): e8732, 2019.
- Cirino-Silva R, Kmit FV, Trentin-Sonoda M, Nakama KK, Panico K, Alvim JM, Dreyer TR, Martinho-Silva H, Carneiro-Ramos MS. Renal ischemia/reperfusion-induced cardiac hypertrop in mice: Cardiac morphological and morphometric characterization. JRSM Cardiovasc Dis. 2017 Jan doi: 10.1177/2048004016689440.
- Trentin-Sonoda M, da Silva RC, Kmit FV, Abrahão MV, Monnerat Cahli G, Brazil GV, Muzi-Filho H, Silva PA, Tovar-Moll FF, Vieyra A, Medei E, Carneiro-Ramos MS. Knockout of Toll-Like Receptors 2 and 4 Prevents Renal Ischemia-Reperfusion-Induced Cardiac Hypertrophy in Mice. PLoS One. 2015 Oct 8; 10 (10): e0139350. DOI: 10.1371/journal. Pone. 0139350.
How was the number of animals per group decided?
Answer: The experimental protocol was evaluated and approved by the ethics committee of the University, which defines, by mathematical formula, the number of animals allowed per protocol. In the present study, several different techniques were performed, which leads to a variation in the number of animals. For example:
- ischemia/reperfusion surgery: the mortality rate is low in this protocol (< 5%) but it is not zero, which means that in some sets of surgeries, we had the loss of some animals from some groups.
- High-resolution respirometry: the extraction of mitochondria is a long and laborious procedure, and we tried to keep the same time between extraction and analysis in Oroborus equipment. However, in some cases, the amount of mitochondria was not enough to carry out the experiment, decreasing the n in certain groups.
Finally, it is important to empathize that for each experiment it was necessary to prepare the samples according to the technique (for example, RT-PCR, NO measurement, etc.) considering the total amount of animals approved by ethics committee. Thus, sometimes, it was not possible to use the same tissue samples for all analysis proposed.
What was the anesthesia method for the mice?
Answer: We would like to thank for the comment regarding our manuscript. The animals were first weighted and submitted to injectable anesthesia according to their weights. The anesthesia was composed by 10mg/kg of xylazine hydrochloride and 80mg/kg of ketamine hydrochloride (both from Agribands do Brasil Ltda, São Paulo). A single dose of 200uL was inject intraperitoneally (IP).
Even if the text mentions that Vit C treatment in the “early group” stops at day 8, it would be helpful to the readers if this is also graphically shown in Fig 1A.
Answer: We would like to thank for the comment regarding our manuscript. We changed the figure 1A due to the better understanding of this point.
Reviewer 3 Report
In this study, the authors investigated the beneficial effects of vitamin C administration on renal and cardiac function in a mice model of cardiorenal syndrome type 3. The results suggest that short-term treatment vitamin C improves renal and mitochondrial function and modulated mitochondrial dynamics and nitric oxide production in heart and kidneys. Some comments and questions regarding these findings are listed below.
- The aim of this study was to evaluate whether vitamin C provides protection to heart and the kidneys in an in vivo CRS type 3 model. Although it is shown that vitamin C is able to decrease serum urea and creatinine, information on renal injury is missing Histological analysis of the kidney (H&E or PAS staining) would provide additional information whether tubular injury is prevented/attenuated by vitamin C treatment. Additionally, heart function was not investigated in this study. Cardiac remodeling is often observed in the IR model. Was echocardiography applied in this model? Otherwise, analysis of the heart on extracellular matrix deposition and cardiac hypertrophy would be valuable.
- Instead of the IR 8d group, why was not chosen to have a control group at 23 days post IR that could be used to directly compare the results of the late treatment group to?
- Line 323-325, ‘the late treatment with vitamin C did not avoid the elevated expression of Drp1’. As there is no real control group at 23 days, it is not possible to assign the elevated Drp1 expression in the late vitamin C group as a direct result of IR injury. In addition, no Drp1 overexpression was found in the IR 8d and 15d group. Is it possible that vitamin C itself has adverse effects on the expression of Drp1?
- A dosage of 57 mg/kg/day vitamin C was used. What is the rationale for this dosage? Additionally, why was a treatment duration of 16 days chosen for the early vitamin C group and 8 days for the late group?
- Were male or female mice used in this study? This should be reported in the Methods.
- The number of mice in each group is different between the different measurements. For some measurements the N=3 in the early vitamin C group. How many mice were included in each group and was any loss of mice observed? This should be reported in the manuscript. It may be informative if the individual values for each animal is displayed in the figures, especially when numbers per group are low.
- In the results, line 180-183, vitamin C prevented the increase in creatinine levels in the early group but not the late group. It is then suggested that creatinine is an acute marker of renal dysfunction. If creatinine levels would be an acute marker, the creatinine levels in the IR 8d would be expected to show an increase, which is not the case. Could the authors further elaborate on why serum urea was decreased in both the early and late group, but creatinine only in the early group?
Author Response
Reviewer 3
In this study, the authors investigated the beneficial effects of vitamin C administration on renal and cardiac function in a mice model of cardiorenal syndrome type 3. The results suggest that short-term treatment vitamin C improves renal and mitochondrial function and modulated mitochondrial dynamics and nitric oxide production in heart and kidneys. Some comments and questions regarding these findings are listed below.
The aim of this study was to evaluate whether vitamin C provides protection to heart and the kidneys in an in vivo CRS type 3 model. Although it is shown that vitamin C is able to decrease serum urea and creatinine, information on renal injury is missing Histological analysis of the kidney (H&E or PAS staining) would provide additional information whether tubular injury is prevented/attenuated by vitamin C treatment.
Answer: We would like to thank for the comment regarding our manuscript. In fact, we agree with the revisor about this point, however the main focus investigation of this study are the mitochondrias and their role in the CRS type 3. Other studies in our laboratory are focusing in renal injury and we will perform the histological analysis also.
Additionally, heart function was not investigated in this study. Cardiac remodeling is often observed in the IR model. Was echocardiography applied in this model? Otherwise, analysis of the heart on extracellular matrix deposition and cardiac hypertrophy would be valuable.
Answer: We would like to thank for the comment regarding our manuscript. Our model of renal IR provides heart injury already observed in other studies. The cardiac hypertrophy is the most studied for us. It was already observed the increasement of heart weight/body weight and heart weight/tibia after 12 and 15 days of reperfusion, respectively. Cardiac hypertrophy markers, B-type natriuretic peptide (BNP) and α-actin, left ventricle mass, cardiac wall thickness and myocyte width after were increased 15 days of reperfusion, together with longer QTc and action potential duration [1]. In our model of renal IR, the echocardiographic has already been done:
We also characterize the collagen content in these groups and the renal ischemia/reperfusion-induced cardiac hypertrophy is not accompanied by fibrosis, evaluated by FTIR-Raman spectroscopy [2].
- Trentin-Sonoda M, da Silva RC, Kmit FV, Abrahão MV, Monnerat Cahli G, Brazil GV, Muzi-Filho H, Silva PA, Tovar-Moll FF, Vieyra A, Medei E, Carneiro-Ramos MS. Knockout of Toll-Like Receptors 2 and 4 Prevents Renal Ischemia-Reperfusion-Induced Cardiac Hypertrophy in Mice. PLoS One. 2015 Oct 8; 10 (10): e0139350. DOI: 10.1371/journal. Pone. 0139350.
- Cirino-Silva R, Kmit FV, Trentin-Sonoda M, Nakama KK, Panico K, Alvim JM, Dreyer TR, Martinho-Silva H, Carneiro-Ramos MS. Renal ischemia/reperfusion-induced cardiac hypertrop in mice: Cardiac morphological and morphometric characterization. JRSM Cardiovasc Dis. 2017 Jan doi: 10.1177/2048004016689440.
Instead of the IR 8d group, why was not chosen to have a control group at 23 days post IR that could be used to directly compare the results of the late treatment group to?
Answer: We would like to thank for the comment. We have already performed the experiments of renal injury evaluation and reperfusion for 2, 4, 6 and 8 weeks and in our lab (data not published). We have supported that the renal injury is maintained until the end of the 8th week compared with Sham 8weeks animals, demonstrating that the treatment with Vit C is important during the AKI protection.
Line 323-325, ‘the late treatment with vitamin C did not avoid the elevated expression of Drp1’. As there is no real control group at 23 days, it is not possible to assign the elevated Drp1 expression in the late vitamin C group as a direct result of IR injury. In addition, no Drp1 overexpression was found in the IR 8d and 15d group. Is it possible that vitamin C itself has adverse effects on the expression of Drp1?
Answer: We would like to thank the reviewer for this observation. Unfortunately, the literature is very limited regarding the modulation of mitochondrial function by vitamin C. Our results, focused on kidney and heart, showed that vitamin C increases the Drp1 expression, in both situations (early and late treatment). However, we don’t know exactly the reason. More experiments are needed to clear this specific point.
A dosage of 57 mg/kg/day vitamin C was used. What is the rationale for this dosage? Additionally, why was a treatment duration of 16 days chosen for the early vitamin C group and 8 days for the late group?
Answer: We would like to thank for the comment regarding our manuscript. The 57mg/kg/day is a well-studied dose that was chosen taking into consideration studies involving renal and heart injuries [1-3]. Regarding the treatment timepoints, the Vit C treatment was thought by the authors in these two specific timepoints given the possibility of a treatment starting in the begging of the syndrome, - simulating a health patient that faces a renal IR due to some acute factors - and also a patient that discovers the AKI and after starts the Vit C treatment. In both cases we have scientific evidences that supported us in these choices.
For the early treatment: Previous studies form our lab have shown that after 8 days of reperfusion we have a strong participation of inflammation and ROS [4-9] and both components have maintained until the day 15 (another super studied timepoint), when we can find renal progression of AKI observed in mRNA levels of vimentin that is a tubular injury marker. In this sense, we have investigated an initial treatment based on known consequences for our model, which Vit C proved effective in preventing.
For the late treatment: Based in the same studies of implementation of renal injury after this model of renal IR, we have thought about an already established injury in the 15th day later treated with Vit C for some more days.
In fact, the observation of control animals in 23 days after reperfusion are primordial, however we have already performed the experiments of renal injury evaluation and reperfusion for 2, 4, 6 and 8 weeks and in our lab (not published). We have supported that the renal injury is maintained until the end of the 8th week compared with Sham 8weeks animals, demonstrating that the treatment with Vit C is important during the AKI protection.
- Zhu, Y.; Zhanga, Y.; Zhanga, J.; Zhanga, Y. Evaluation of Vitamin C Supplementation on Kidney Function and Vascular Reactivity Following Renal Ischemic Injury in Mice. Kidney Blood Press Res. 2016, 41, 460–470, doi:10.1159/000443447.
- Bell, J.P.; Mosfer, S.I.; Lang, D.; Donaldson, F.; Lewis, M.J. Vitamin C and quinapril abrogate LVH and endothelial dysfunction in aortic-banded guinea pigs. Am J Physiol Hear. Circ Physiol 2001, 281, H1704–H1710, doi:10.1152/ajpheart.2001.281.4.H1704.
- Koul, V.; Kaur, A.; Singh, A.P. Investigation of the role of nitric oxide/soluble guanylyl cyclase pathway in ascorbic acid-mediated protection against acute kidney injury in rats. Mol Cell Biochem. 2015, 406, 1–7, doi:10.1007/s11010-015-2392-4.
- Silva W.C., Dias D.S., Junho C.V.C., Panico K., Santos R.N., Pelegrino M.T., Pieretti J.C., Seabra A.B., Angelis K., Carneiro-Ramos M.S. Characterization of the oxidative stress in renal ischemia/reperfusion-induced cardiorenal syndrome type 3. Biomed Research International, v. 2020, p. 1605358, 2020.
- Panico K, Abrahão MV, Trentin-Sonoda M, Muzi-Filho H, Vieyra A, Carneiro-Ramos MS. Cardiac Inflammation after ischemia-reperfusion of the kidney: Role of the sympathetic nervous system and the renin-angiotensin system. Cell Physiol Biochem;53(4):587-605. doi: 10.3594/000000159, 2019.
- Alarcon MML, Trentin-Sonoda M, Panico K, Schleier Y, Duque T, Moreno-Loaiza O, de Yurre AR, Ferreira F, Caio-Silva W, Coury PR, Paiva CN, Medei E, Carneiro-Ramos MS. Cardiac arrhythmias after renal I/R depend ON IL-1β, Journal of Molecular and Cellular Cardiology, 25. PII: S0022-2828 (19) 30083-5, 2019.
- Junho CV, Trentin-Sonoda M, Alvim JM, Gaisler-Silva J, Carneiro-Ramos MS. TLR4-induced cardiomyocyte hypertrophy is dependent of camkii through complement system and NF-KB. Brazilian Journal of Medical and Biological Research, 52 (7): e8732, 2019.
- Cirino-Silva R, Kmit FV, Trentin-Sonoda M, Nakama KK, Panico K, Alvim JM, Dreyer TR, Martinho-Silva H, Carneiro-Ramos MS. Renal ischemia/reperfusion-induced cardiac hypertrop in mice: Cardiac morphological and morphometric characterization. JRSM Cardiovasc Dis. 2017 Jan doi: 10.1177/2048004016689440.
- Trentin-Sonoda M, da Silva RC, Kmit FV, Abrahão MV, Monnerat Cahli G, Brazil GV, Muzi-Filho H, Silva PA, Tovar-Moll FF, Vieyra A, Medei E, Carneiro-Ramos MS. Knockout of Toll-Like Receptors 2 and 4 Prevents Renal Ischemia-Reperfusion-Induced Cardiac Hypertrophy in Mice. PLoS One. 2015 Oct 8; 10 (10): e0139350. DOI: 10.1371/journal. Pone. 0139350
Were male or female mice used in this study? This should be reported in the Methods.
Answer: We only used Male C57BL6 mice. We added this information in the manuscript.
The number of mice in each group is different between the different measurements. For some measurements the N=3 in the early vitamin C group. How many mice were included in each group and was any loss of mice observed? This should be reported in the manuscript. It may be informative if the individual values for each animal is displayed in the figures, especially when numbers per group are low.
Answer: We would like to thank for the comment. The experimental protocol was evaluated and approved by the ethics committee of the University, which defines, by mathematical formula, the number of animals allowed per protocol. In the present study, several different techniques were performed, which leads to a variation in the number of animals. For example:
- ischemia/reperfusion surgery: the mortality rate is low in this protocol (< 5%) but it is not zero, which means that in some sets of surgeries, we had the loss of some animals from some groups.
- High-resolution respirometry: the extraction of mitochondria is a long and laborious procedure, and we tried to keep the same time between extraction and analysis in Oroborus equipment. However, in some cases, the amount of mitochondria was not enough to carry out the experiment, decreasing the n in certain groups.
Finally, it is important to empathize that for each experiment it was necessary to prepare the samples according to the technique (for example, RT-PCR, NO measurement, etc.) considering the total amount of animals approved by ethics commitee.
In the results, line 180-183, vitamin C prevented the increase in creatinine levels in the early group but not the late group. It is then suggested that creatinine is an acute marker of renal dysfunction. If creatinine levels would be an acute marker, the creatinine levels in the IR 8d would be expected to show an increase, which is not the case. Could the authors further elaborate on why serum urea was decreased in both the early and late group, but creatinine only in the early group?
Answer: We would like to thank for the comment. It is known that the levels of creatinine and urea are increased in AKI, and also are considered a biomarkers of kidney injury, although creatine seems to be more specific marker [1, 2]. In relation to our results, it’s possible that the early treatment with vitamin C is more effective to prevent a damage in renal function instead the late treatment, when its possible to have a strong injury in renal tissue. We would like to perform other experiments showing, for example, glomerular filtration rate, and confirm this data. Unfortunately, it’s not possible at this time.
- Edelstein, C.L. Biomarkers of Acute Kidney Injury. Adv Chronic Kidney Dis 2008, 15, 222–234, doi:10.1053/j.ackd.2008.04.003.
- Sidebotham, D. Novel Biomarkers for Cardiac Surgery-Associated Acute Kidney Injury: A Skeptical Assessment of Their Role. JECT 2012, 44, 235–240.
Round 2
Reviewer 2 Report
I understand that the authors have evaluated the renal and cardiac injury observed after 23 days of IR in previous studies, however, since no control group at day 23 was included for this specific study, the comparisons for the late VitC group are indirect and the authors should add a sentence on this topic to the limitation section.
Author Response
Reviewer 2
I understand that the authors have evaluated the renal and cardiac injury observed after 23 days of IR in previous studies, however, since no control group at day 23 was included for this specific study, the comparisons for the late VitC group are indirect and the authors should add a sentence on this topic to the limitation section.
Answer: We would like to thank for the comment regarding our manuscript. We’ve included a sentence in the limitation section.
Reviewer 3 Report
I thank the authors for their responses to my questions. Below are two more recommendations for the manuscript.
- As the authors explained that the main focus of this study is on the mitochondria and their role in CRS type 3, I would strongly suggest to revise the title of the manuscript: ‘Mitochondrial dysfunction in Cardiorenal Syndrome 3: reno- cardiac protection of Vitamin C’. The second part of the title implies that vitamin C treatment has beneficial effects on renal and cardiac function, while this is not investigated and might be a misleading to the reader.
- Due to the large variation in animal numbers between the different techniques used (from N=4 to 15), I would highly suggest to show all the data as dot plots instead of as bar graphs only, as this provides more information and is more transparent.
Author Response
Reviewer 3
I thank the authors for their responses to my questions. Below are two more recommendations for the manuscript.
As the authors explained that the main focus of this study is on the mitochondria and their role in CRS type 3, I would strongly suggest to revise the title of the manuscript: ‘Mitochondrial dysfunction in Cardiorenal Syndrome 3: reno- cardiac protection of Vitamin C’. The second part of the title implies that vitamin C treatment has beneficial effects on renal and cardiac function, while this is not investigated and might be a misleading to the reader.
Answer: We would like to thank for the suggestion, and we’ve changed the title of the manuscript. The new title is: Mitochondrial dysfunction in Cardiorenal Syndrome 3: renocardiac effect of Vitamin C
Due to the large variation in animal numbers between the different techniques used (from N=4 to 15), I would highly suggest to show all the data as dot plots instead of as bar graphs only, as this provides more information and is more transparent.
Answer: We would like to thank for the suggestion and we’ve altered the graphs style to provide more information and transparency.